# Photosynthetic Responses of Two Woody Halophyte Species to Saline Groundwater Irrigation in the Taklimakan Desert

**Jiao Liu** [1,2], **Ying Zhao** [1,2,*], **Tanveer Ali Sial** [1,3], **Haidong Liu** [2], **Yongdong Wang** [2] **and Jianguo Zhang** [1,*]

1   Key Laboratory of Plant Nutrition and the Agri-Environment in Northwest China, Ministry of Agriculture, Northwest A&F University, Xianyang 712100, China; liuxy0803@163.com (J.L.); alisial@nwafu.edu.cn (T.A.S.)
2   Xinjiang Institute of Ecology and Geography, Chinese Academy of Sciences, Urumqi 830011, China; liu_hducas@163.com (H.L.); wangyd@ms.xjb.ac.cn (Y.W.)
3   Department of Soil Science, Sindh Agriculture University, Tandojam 70060, Pakistan
*   Correspondence: yzhaosoils@gmail.com (Y.Z.); zhangjianguo21@nwafu.edu.cn (J.Z.)

**Abstract:** The study of plant photosynthesis under different degrees of drought stress can provide a deeper understanding of the mechanism of plant drought resistance. In the Taklimakan Desert, saline groundwater is the only local water source with regard to shelterbelt construction and determines plant growth and photosynthetic changes. In this study, daily dynamics of gas exchange parameters and their responses to photosynthetic photon flux density at three irrigation levels (W1 = 17.5, W2 = 25, W3 = 35 mm) were measured for two main species, i.e., *Calligonum mongolicum* (*C*) and *Haloxylon ammodendron* (*H*). *H* was better adapted to drought stress than *C*. Net photosynthetic rate ($P_N$) was mainly related to soil water status in the main root system activity layer. In July, the daily variations of $P_N$ and transpiration ($T_r$) for *C* were higher than *H*. *C* increased water use efficiency (WUE) with increases in $P_N$, while *H* decreased $T_r$ to obtain a higher WUE. Either *C* or *H*, drought reduced the low light and metabolic capacity, and thus decreased the light adaptability and photosynthesis potential. We suggest a prerequisite understanding of physiological mechanisms and possible plant morphological adjustments required to adapt plant species to desert drought conditions.

**Keywords:** desert plant; photosynthesis; drought stress; drip irrigation

## 1. Introduction

Desertification is a global concern and vegetation restoration has been regarded as an effective ecological measure for desertification control. However, water deficiency is one of the major limitations that affect the physiological processes and ecological adaptability of plants in arid regions of the world [1,2]. Desert plants grow in a harsh environment with high temperatures and are normally exposed to drought, which seriously influences the distribution and growth of plants. However, desert plants habitually reduce water loss and alleviate high-radiation damage to their photosynthetic apparatus through specific morphological and physiological features [3]. It is well known that desert plants can tolerate adversity and adapt at different growth stages. There are also significant differences between plant species. It is crucial to develop a better understanding of the adaptation capabilities of plants to their environments and find ways to facilitate those adaptations.

Photosynthesis is sensitive to environmental changes and may be used to estimate plant adaptation to their habitats. As the basic unit of physiological metabolism and matter accumulation, photosynthesis is an important link to plant growth and metabolism [4,5]. Photosynthetic tissue adapts to high external radiation by protecting the assimilation process through diurnal adjustments in photochemical and non-photochemical processes [6]. Numerous studies have indicated that desert shrubs have evolved unique physiological and morphological characteristics in adapting to the harsh environment and exhibit a higher tolerance to water shortage than other plants [7]. However, their metabolic activities are still vulnerable to variations in water availability [8–10]. This vulnerability is

due to the main physiological parameters of desert plants, such as stomatal conductance, photosynthetic rate and transpiration rate, which are variably inhibited during times of water shortage [11–15].

The Taklimakan Desert is located in northwest China and is the second-largest shifting sand desert in the world. It is an extremely harsh environment where most areas are covered by shifting sand. The Taklimakan Desert Highway Shelterbelt was constructed in 2003 to protect the highway and is irrigated with underground water. Three main plant species are grown in the shelterbelt, *Haloxylon ammodendron* (*H*), *Calligonum mongolicum* (*C*) and *Tamarix ramosissima*. These plant species are highly salt-tolerant and drought-resistant with excellent windbreak and sand fixation properties. However, it has long been speculated that these plant species may suffer from water and salt stresses [16]. The physiological characteristics of *C* and *H* and their photosynthetic organs conform with Kranz anatomy and are typical C4 plants with carbon isotopic ratios ($\delta^{13}$C) of $-15\%$, $CO_2$ compensation points of less than 5 $\mu$mol mol$^{-1}$, light saturation points (LSP) of higher than 1600 $\mu$mol m$^{-2}$ s$^{-1}$ and elevated photosynthetic capacities combined with high water use efficiencies [17]. Current concerns about the relationship between plant-available water and plants under drought stress focus mainly on the photosynthesis of natural vegetation in dry climates [17–20]. There is limited research on the photosynthetic characteristics of artificial forests under saltwater irrigation.

The primary problem encountered when trying to establish and support vegetation in desert ecosystems is the availability of good quality water. In the Taklimakan Desert, groundwater is the only local water source with regard to shelterbelt construction. However, the solutes in groundwater may increase soil salt contents and affect plant growth and survival in irrigated areas [21]. When exposed to salt stress, plant growth may be influenced by the osmotic effects on water uptake. The irrigation levels influence physiological activity, individual morphology and even the long-term adaptive strategies of desert plants. Under these circumstances, we tested how plants adapt to different soil moisture statuses and the impacts of different irrigation levels on *C* and *H*. We considered that the plant physiological response to irrigation levels should be closely related to soil moisture contents and that there may be variation between different species. Our objectives were to investigate: (1) the effects of irrigation levels on the plants and (2) the photosynthesis adaptations of two plant species in response to drought stress and, more specifically, the differences between them. Only when careful choices are made concerning the salinity of available irrigation water and the salinity tolerance of appropriate plant species can vegetation restoration promise sustainable development.

## 2. Materials and Methods

### 2.1. Environmental Conditions of the Study Area

The study was conducted at the Taklimakan Desert Research Station, Chinese Academy of Sciences (CAS). The station is located in the middle of the Taklimakan Desert (Tazhong), China (39°00′ N and 83°40′ E) at an altitude of 1099 m (Figure 1). The climate is an arid desert climate with an average annual precipitation of <50 mm and average annual evaporation of >3000 mm [22]. According to the Tazhong weather station's records, the average temperature is 12.4 °C, but the maximum and minimum temperatures may reach 45.6 °C and $-22.2$ °C, respectively. The soil is aeolian sand with low nutrient contents, poor moisture capacity and limited fertilizer retention. The salt content ranges from 1.26 to 1.63 g kg$^{-1}$, and the pH ranges from 8 to 9 [23]. The Taklimakan Desert Forest is an artificial plantation where *H. ammodendron* and *C. mongolicum* are the main tree species.

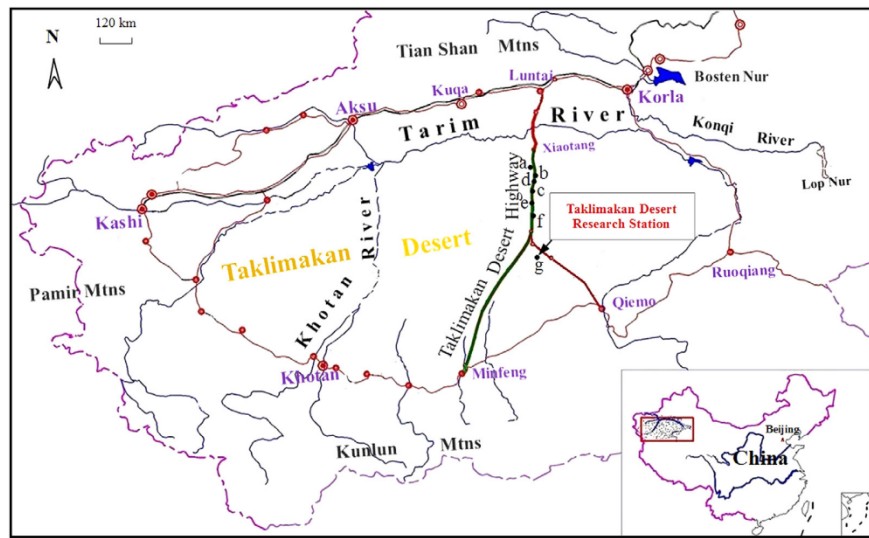

**Figure 1.** Taklimakan Desert Highway and the monitoring sites.

### 2.2. Difference in Irrigation Management

We selected six rows of *H* and *C* in the current study in 2016. All plants were the same age (8 years old) and were planted in the same field. Each row had one type of plant and the interval between adjacent rows was greater than 5 m to avoid water exchange between the different rows. Twenty trees were planted in each row and the distance between each tree was 1 m. Drip irrigation pipes were laid close to the trunk, the water outlet holes were spaced with 1 m, and the drip flow rate was 1.5 L/h. The specific field configuration is shown in Figure 2. All *H* and *C* trees were irrigated with saline groundwater with a 10-day time interval through a drip irrigation system with three irrigation levels of 17.5 mm (W1), 25 mm (W2) and 35 mm (W3), and all other basic agronomical practices were the same. In September and October, the time interval of irrigation increased so that plants were irrigated with groundwater every 15 days. Plants were not irrigated from November to February. The salt concentration of the groundwater was 4.02 g L$^{-1}$.

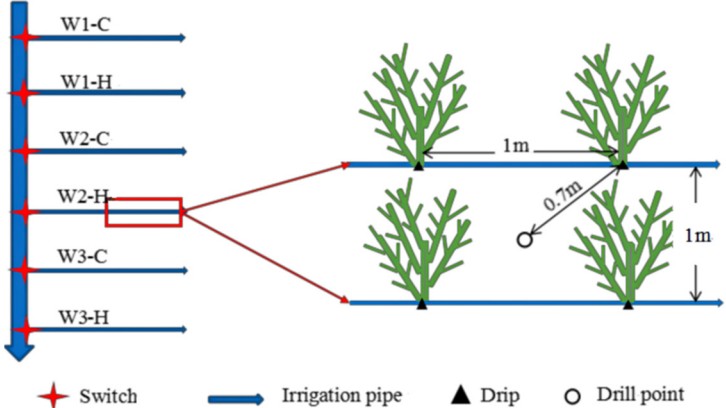

**Figure 2.** Schematic diagram of field experiment design.

### 2.3. Soil Physical and Chemical Properties

The soil moisture content was measured using the oven-dry method. Three soil samples were collected from each plot to determine the soil water content using oven-drying and weighing methods. Each soil sample was collected at a depth of 0–200 cm during photosynthesis measurement.

The soil was shifting Aeolian sand with a sandy texture. The salt content was 1.26 g kg$^{-1}$ and it had a slightly alkaline reaction with a pH value of 7.69, a bulk soil

density of 1.63 g cm$^{-3}$, field capacity of 21.27%, a soil organic matter content of 0.94 g kg$^{-1}$, a total soil nitrogen content of 0.03 g kg$^{-1}$, a total soil phosphorus content of 0.40 g kg$^{-1}$ and a total soil potassium content of 16.62 g kg$^{-1}$.

### 2.4. Photosynthesis Measurements

The daily gas exchange dynamics and the responses to photosynthetic photon flux density (PPFD) were measured on typically sunny days during the same growth stage using an LI–6400 Portable Photosynthetic System (LI-COR, Lincoln, NE, USA). To investigate the drought-stress responses of the two shrub species when combined with high temperatures, we measured the daily gas exchange dynamics each day in July (Beijing time 9:00–20:00), and took measurements at intervals of two hours in each leaf. Sunnyside and undamaged assimilating shoots blade at the middle height of trees were chosen to measure. Four assimilating shoots were selected in each treatment, which were recorded five times for each assimilating shoot. The measurements were made when the sun was directly overhead to have a consistent sun angle. To keep the leaf chambers of the assimilating shoots closed during measurement, we marked the assimilating shoots with colored tape [17]. The same assimilating shoots were measured all day.

Photosynthetic photon flux density (PPFD) was measured using an LI–6400 Portable Photosynthetic System (LI–COR, Lincoln, NE, USA) from July to September to compare seasonal differences in 2016. According to the high temperature and strong radiation in the desert, we set a constant temperature of 30 °C, and the reference $CO_2$ concentration was equal to the environmental $CO_2$ concentration (380 mmol mol$^{-1}$). The light source was provided via an RGB Light, and the intensity ranged from 2200 to 0 mol m$^{-2}$ s$^{-1}$ for which the gradient gradually decreases in 12 grades. The data were recorded with the instrument's automatic observation programs, and each measurement was finished once.

After measuring photosynthesis, the leaves were taken off, and we took photos to calculate their areas with Photoshop, and then the photosynthetic parameters were recalculated.

The light compensation point (LCP) and apparent quantum yield (AQE) were obtained by fitting a linear regression of net photosynthetic rate ($P_N$) against PPFD ($\leq$200 µmol m$^{-2}$ s$^{-1}$). The light saturation point (LSP), dark respiration rate ($R_d$) and maximum net photosynthetic rate ($P_{max}$) were obtained by fitting a Farqhuar non-linear hyperbolic model of $P_N$ against PPFD ($\geq$200 µmol m$^{-2}$ s$^{-1}$) [3,24].

$$P_N = \frac{\Phi * Q + P_{max} - \sqrt{(\Phi * Q + P_{max})^2 - 4 * \Phi * Q * k * P_{max}}}{2k} - R_d \tag{1}$$

where $P_N$ is the net photosynthetic rate; $\Phi$ is apparent quantum yield; $Q$ is effective photosynthetic radiation incident to the leaf; $P_{max}$ is the maximum net photosynthetic rate; $R_d$ is dark respiration rate; and $k$ is the curved angle of the light response curve. φ

Plant water use efficiency (i.e., instantaneous; $CO_2$ assimilation/transpiration) is calculated as:

$$WUE = P_N/E \tag{2}$$

$P_N$ is the net photosynthetic rate (µmol $CO_2$), and $E$ is transpiration (mol $H_2O$).

### 2.5. Statistical Analyses

The statistical analyses were conducted using IBM SPSS Statistics (19.0, Software, USA) and figures were created using Origin 2016. The results for the parameters are presented as t-tests arranged according to photosynthetic rates under the different irrigation levels of the two woody plant species using IBM SPSS Statistics software 19.0. Multiple comparisons of various levels were performed using Duncan's New Multiple Range Test.

## 3. Results

### 3.1. Soil Moisture Content

Soil water content varied greatly between months. Overall, the water content of the shallow soil layer (0–100 cm) in July, August and September was greater than that of the deep soil layer (100–200 cm). The maximum soil water content was 40–60 cm, the average soil water content of 0–200 cm in July was the lowest, and that in September was the highest.

Differences between months of the same treatment were tested by paired-samples T test, which showed that all treated soil water content was insignificant between August and September, while differences between July and August and between July and September reached significant or extremely significant levels ($p < 0.01$).

In July, August and September, the soil water content of 0–100 cm layer under W2 treatment was less than W3 and W1 for *C*, and the content under W3 was the largest for *H*. The soil water content under W2 treatment was the lowest in July and August, and the soil water content under W1 treatment was the lowest in September for *H* and *C* (Figure 3).

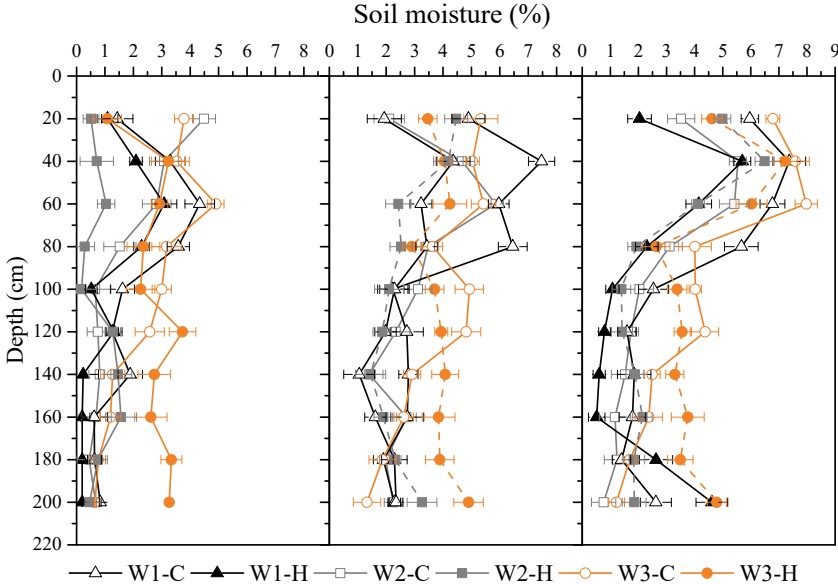

**Figure 3.** The soil moisture content in the low (W1), medium (W2) and high (W3) irrigation amounts during July, August and September of 2016. The C and H present *C. mongolicum* and *H. ammodendron*, respectively, same in the figures blow.

### 3.2. Daily Changes in the Meteorological Factors for the Different Months

Figure 4 indicated that the photosynthetic photon flux density (PPFD) increased antemeridian, reaching the highest value at noon, maintaining the maximum value until 15:30, and declining until sunset. The maximum PPFD was recorded in July, where the whole day maximum was 2112 µmol m$^{-2}$ s$^{-1}$ and was 1825 µmol m$^{-2}$ s$^{-1}$ in September. There was a trend of increasing air temperatures (T$_a$) throughout the experiment. A higher T$_a$ was recorded in July than in August and September. A T$_a$ greater than 30 °C was recorded after 10:30 and the maximum value was recorded at 17:30 at 40.1 °C in July. The mean T$_a$ values in August and September between 8:30 and 20:00 were 25.58 °C and 24.54 °C, respectively, and there was no significant difference between the two months after 11:30.

The relative humidity (RH) declined from 8:30 to 20:00 during the day. The highest values were recorded in August and the lowest in July, and the mean values for July, August and September between 8:30 and 20:00 were 12.15%, 49.67% and 27.11%, respectively.

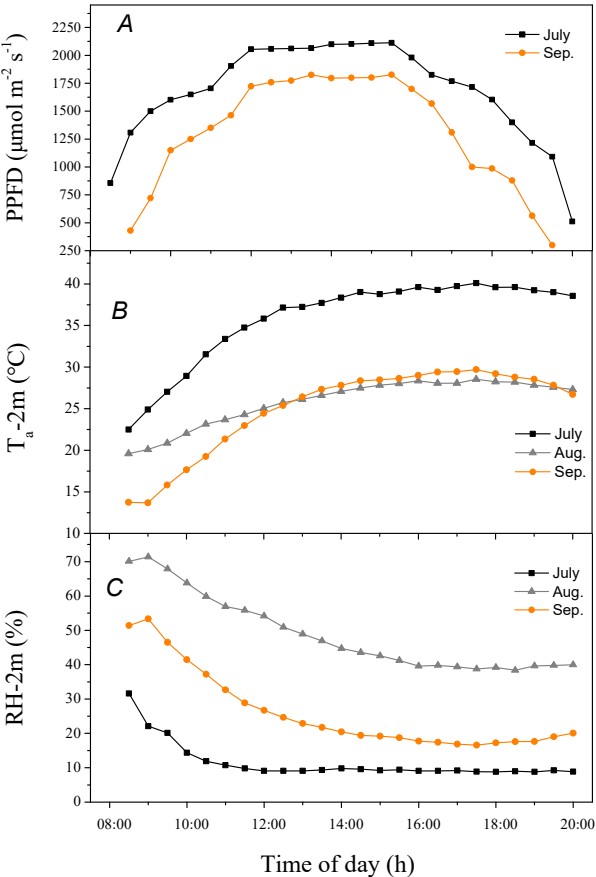

**Figure 4.** Diurnal changes in the photon flux density (PPFD) (**A**), air temperature (Ta) (**B**) and air relative humidity (RH) (**C**) for July, August and September in 2016.

*3.3. Diurnal Changes of the Photosynthetic Rate under the Different Irrigation Levels*

At the plant growth stage, we measured plant height and crown width under different treatments, and the analysis showed that there was no significant difference in the increase of plant height and crown width among all treatments. Different irrigation treatments had limited effects on plant height and crown width.

The diurnal differences in the net photosynthetic rates ($P_N$) of *C* and *H* under different irrigation levels are shown in Figure 5. The daily variations of $P_N$ for *C* in the W1 treatment showed a bimodal curve and displayed an obvious "midday depression" phenomenon. However, only a slight "midday depression" phenomenon was observed in the W2 and W3 treatments. The depression of $P_N$ for *C* in the W1, W2 and W3 treatments appeared around 14:00. For the daily mean $P_N$ (from 9:00–19:00), the highest value was recorded in W1 (21.27 μmol ($CO_2$) m$^{-2}$ s$^{-1}$), followed by the W3 (14.86 μmol ($CO_2$) m$^{-2}$ s$^{-1}$) and W2 (13.80 μmol ($CO_2$) m$^{-2}$ s$^{-1}$) treatments.

In contrast, $P_N$ was generally lower in *H* than *C*. The daily variations of $P_N$ for *H* in the W1 treatment showed a unimodal pattern with a maximum value around noon. The $P_N$ values in the W2 and W3 treatments displayed a bimodal curve. A depression occurred around 13:00 h in the W2 treatment and around 15:00 h in the W3 treatment. The daily mean $P_N$ (from 9:00 to 19:00) was highest in the W3 treatment with a maximum value of 11.61 μmol ($CO_2$) m$^{-2}$ s$^{-1}$, followed by the W1 (7.85 μmol ($CO_2$) m$^{-2}$ s$^{-1}$) treatment with a maximum value 10.25 μmol ($CO_2$) m$^{-2}$ s$^{-1}$ and the W2 (6.56 μmol ($CO_2$) m$^{-2}$ s$^{-1}$) treatment with maximum value 8.98 μmol ($CO_2$) m$^{-2}$ s$^{-1}$. The daily mean $P_N$ of *C* was significantly higher in the W3 treatment than in the W2 treatment ($p < 0.05$). There were no statistically significant differences in the daily mean $P_N$ values between the W1 and W2 treatments or between the W1 and W3 treatments ($p > 0.05$).

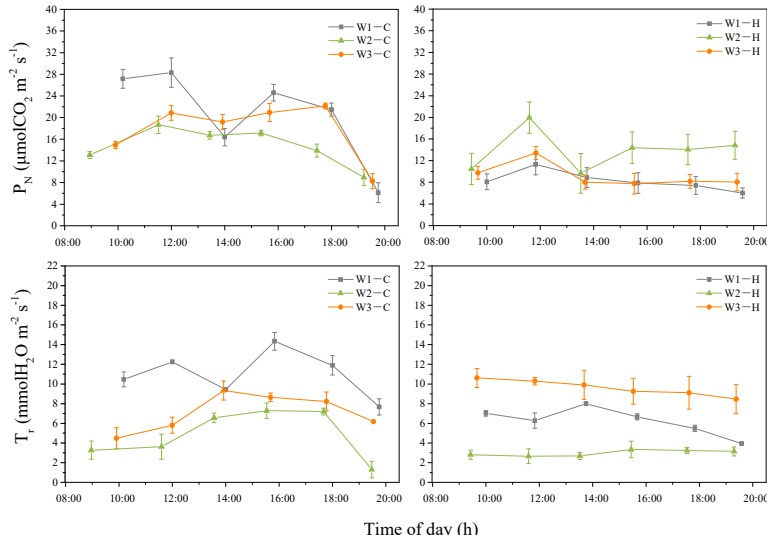

**Figure 5.** Diurnal changes in the net photosynthetic rate ($P_N$) and transpiration ($T_r$) of *C. mongolicum* (*C*) and *H. ammodendron* (*H*) in the low (W1), medium (W2) and high (W3) irrigation amounts.

Transpiration ($T_r$) for the different irrigation levels generally exhibited a similar pattern to net photosynthesis (Figure 3). The diurnal variations of $T_r$ for *C* were unimodal and reached a peak at 13:30 h in the W2 and W3 treatments. On the contrary, the $T_r$ values displayed a bimodal distribution in the W1 treatment and reached peaks at 12:00 and 16:00 h. For daily mean $T_r$ (from 9:00 to 19:00) the highest values were obtained in the W1 treatment (13.25 mmol ($H_2O$) $m^{-2}$ $s^{-1}$ and 16.51 mmol ($H_2O$) $m^{-2}$ $s^{-1}$), followed by the W3 treatment (8.97 mmol ($H_2O$) $m^{-2}$ $s^{-1}$ and 11.03 mmol ($H_2O$) $m^{-2}$ $s^{-1}$) and the W2 treatment (7.24 mmol ($H_2O$) $m^{-2}$ $s^{-1}$ and 9.34 mmol ($H_2O$) $m^{-2}$ $s^{-1}$. The diurnal variations of $T_r$ for *H. ammodendron* were unimodal and reached a peak at 15:00 h in the W1 treatment. The $T_r$ values decreased before 11:30 am and increased to reach a peak at 14:00 until sunset in the W2 and W3 treatments. The daily mean highest $T_r$ was recorded from 9:00 to 19:00 in the W1 treatment (6.23 mmol ($H_2O$) $m^{-2}$ $s^{-1}$), followed by the W3 (5.24 mmol ($H_2O$) $m^{-2}$ $s^{-1}$) and W2 (2.93 mmol ($H_2O$) $m^{-2}$ $s^{-1}$) treatments.

In all treatments, the diurnal changes of intercellular $CO_2$ concentration ($C_i$) of *C* and *H* exhibited a W-shaped response curve (Figure 6). The diurnal changes of $L_S$ were opposite to that of $C_i$.

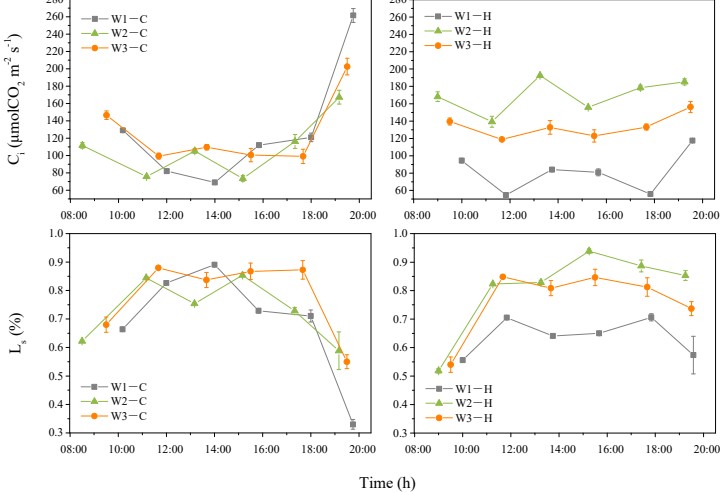

**Figure 6.** Differences in the intercellular $CO_2$ concentrations ($C_i$) and stomatal limitations ($L_s$) of *C. mongolicum* (*C*) and *H. ammodendron* (*H*) in the low (W1), medium (W2) and high (W3) irrigation amounts.

The factors influencing the decline of net photosynthetic rate of plants can be divided into stomatal limiting factors and non-stomatal limiting factors. $C_i$ decreased and Ls increased when $P_N$ decreased in W1-C treatment from 12:00 to 14:00, W1-H treatment from 14:00 to 18:00 and W3-H treatment from 14:00 to 16:00, indicating controls by stomatal limiting factors. In contrast, during other periods, $P_N$ decreases were controlled by non-stomatal limiting factors.

### 3.4. The Response of $P_N$ to Photosynthetic Photon Flux Density (PPFD) under Different Irrigation Levels

The response of $P_N$ to photosynthetic photon flux density (PPFD) under different irrigation levels during July, August and September is presented in Figure 7. The physiological parameters are further calculated according to the response curve of $P_N$ to photosynthetic photon flux density (PPFD), which reflects the photosynthetic capacities of the plant (Table 1).

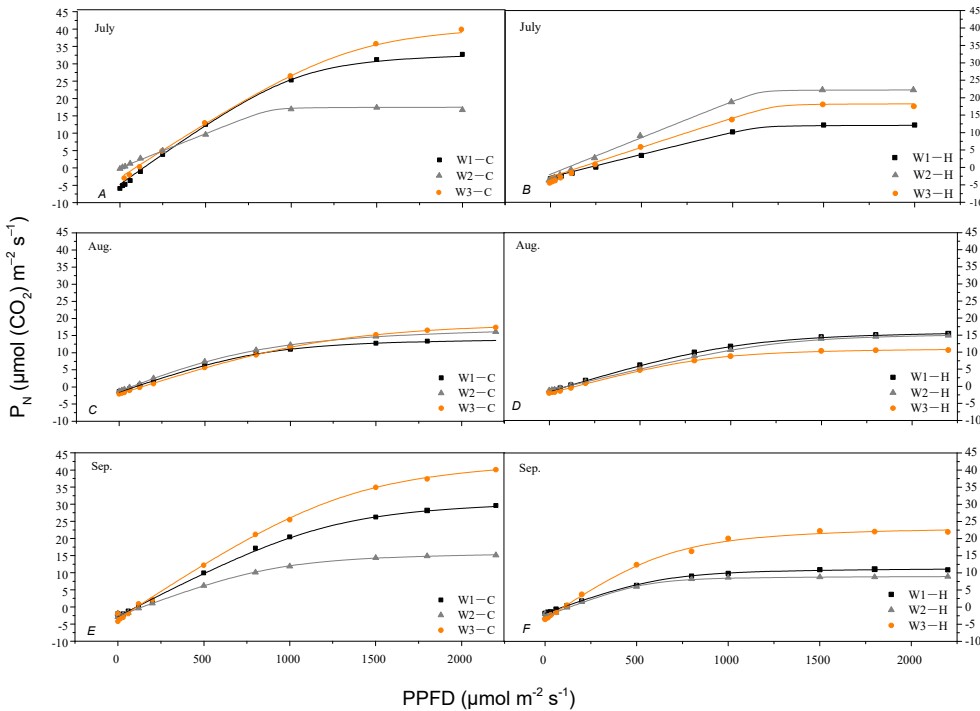

**Figure 7.** The photosynthetic light response curves ($P_N$) to photosynthetic photon flux density (PPFD) of *C. mongolicum* (C) and *H. ammodendron* (H) in the low (W1), medium (W2) and high (W3) irrigation amounts in July (**A**,**B**), August (**C**,**D**) and September (**E**,**F**).

The plant species obviously reacted differently during July, August and September due to the different responses of $P_N$ to photosynthetic photon flux density (PPFD) with three irrigation levels. The photosynthesis of *C* irrigated with three irrigation levels was higher than *H* in July and September. However, in August, they were approximately equal in *C* and *H*. The light compensation point (LCP) values of *C* and *H* for the W2 treatment were lowest in July and August and highest in September. In comparing the W1 and W2 treatments, the light saturation point (LSP) of *C* in the W3 treatment was the highest in July and August. However, the LSP values of *C* and *H* were lowest in the W2 treatment irrigated with three irrigation levels in September. The light-saturated rate of $CO_2$ assimilation ($P_{max}$) and dark respiration rate ($R_d$) of *C* in the W3 treatment was obviously the highest values for those parameters during the three months. However, the $P_{max}$ and $R_d$ of *H* in the W3 treatment were higher than in the W1 and W2 treatments in September. The bivariate correlation test results compare the environmental variables and plant physiology characteristics in Table 2.

Plant photosynthesis is mainly related to PPFD and plant auto-regulatory mechanisms, but not to temperature and relative humidity.

**Table 1.** The light compensation point (LCP), light saturation point (LSP), the apparent quantum yield of $CO_2$ assimilation ($\Phi$), dark respiration rate ($R_d$) and light-saturated rate of $CO_2$ assimilation ($P_{max}$) of *C* and *H* under the low (W1), medium (W2) and high (W3) irrigation amounts.

| | LCP [$\mu$mol m$^{-2}$ s$^{-1}$] | | | LSP [$\mu$mol m$^{-2}$ s$^{-1}$] | | | $\Phi$ [mol mol$^{-1}$] | | | $R_d$ [$\mu$mol m$^{-2}$ s$^{-1}$] | | | $P_{max}$ [$\mu$mol($CO_2$) m$^{-2}$ s$^{-1}$] | | |
|---|---|---|---|---|---|---|---|---|---|---|---|---|---|---|---|
| **Month** | **July** | **Aug.** | **Sep.** | **July** | **Aug.** | **Sep.** | **July** | **Aug.** | **Sep.** | **July** | **Aug.** | **Sep.** | **July** | **Aug.** | **Sep.** |
| W1-C | 146.87 | 90.68 | 108.65 | 1113.77 | 1131.84 | 1497.70 | 0.040 | 0.016 | 0.025 | 4.714 | 1.510 | 2.739 | 38.78 | 16.14 | 35.00 |
| W2-C | 49.52 | 71.01 | 138.94 | 1109.12 | 1195.87 | 1043.97 | 0.023 | 0.018 | 0.021 | 1.511 | 1.344 | 2.661 | 23.84 | 19.91 | 19.10 |
| W3-C | 214.07 | 128.94 | 100.56 | 1482.02 | 1528.18 | 1252.68 | 0.040 | 0.016 | 0.042 | 7.339 | 2.041 | 3.583 | 50.72 | 22.11 | 48.50 |
| W1-H | 244.13 | 97.25 | 95.74 | 1869.86 | 1213.77 | 811.68 | 0.013 | 0.016 | 0.019 | 6.036 | 1.574 | 1.804 | 21.78 | 18.31 | 13.39 |
| W2-H | 130.17 | 97.09 | 123.69 | 1096.40 | 1371.49 | 618.06 | 0.025 | 0.013 | 0.024 | 1.744 | 1.302 | 2.624 | 24.35 | 17.08 | 11.77 |
| W3-H | 169.02 | 163.86 | 100.51 | 1082.83 | 1281.61 | 871.18 | 0.023 | 0.012 | 0.036 | 2.615 | 2.038 | 3.570 | 21.11 | 13.41 | 27.67 |

**Table 2.** The bivariate correlation test results comparing the environmental variables and plant physiology characteristics ($P_N$—Net photosynthetic rate; $T_r$—Transpiration rate; $C_i$—Intercellular $CO_2$ concentration; $L_s$—stomatal limitation; $g_s$—Stomatal conductance; PPFD—Photosynthetic photo flux density; $T_a$—Air temperature; RH—Relative humidity; WUE—Water use efficiency; $T_l$—Leaf temperature; *C*—*C. mongolicum*; *H*—*H. ammodendron*) during July 2016.

| | Plant Species | $P_N$ | $T_r$ | $C_i$ | $L_s$ | $g_s$ | PPFD | $T_a$ | RH | WUE | $T_l$ |
|---|---|---|---|---|---|---|---|---|---|---|---|
| $P_N$ | *C* | 1.00 | | | | | | | | | |
| | *H* | 1.00 | | | | | | | | | |
| $T_r$ | *C* | 0.76 ** | 1.00 | | | | | | | | |
| | *H* | 0.80 ** | 1.00 | | | | | | | | |
| $C_i$ | *C* | −0.66 ** | −0.25 | 1.00 | | | | | | | |
| | *H* | −0.34 | −0.45 | 1.00 | | | | | | | |
| $L_s$ | *C* | 0.65 ** | 0.24 | −0.99 ** | 1.00 | | | | | | |
| | *H* | 0.34 | 0.45 | −.99 ** | 1.00 | | | | | | |
| $g_s$ | *C* | 0.69 ** | 0.72 ** | −0.23 | 0.21 | 1.00 | | | | | |
| | *H* | 0.76 ** | 0.80 ** | −0.44 | 0.44 | 1.00 | | | | | |
| PPFD | *C* | 0.66 ** | 0.51* | −0.69 ** | 0.69 ** | 0.13 | 1.00 | | | | |
| | *H* | 0.29 | 0.22 | −0.34 | 0. 34 | −0.04 | 1.00 | | | | |
| $T_a$ | *C* | 0.04 | 0.20 | −0.05 | 0.08 | −0.44 | 0.42 | 1.00 | | | |
| | *H* | −0.10 | −0.07 | −0.19 | 0.21 | −0.37 | 0.49 ** | 1.00 | | | |
| RH | *C* | −0.30 | −0.31 | 0.15 | −0.17 | 0.25 | −0.54 * | −0.91 ** | 1.00 | | |
| | *H* | −0.01 | 0.06 | 0.28 | −0.29 | 0.35 | −0.55 * | −0.94 ** | 1.00 | | |
| WUE | *C* | 0.55 * | −0.09 | −0.73 ** | 0.73 ** | 0.12 | 0.41 | −0.13 | −0.13 | 1.00 | |
| | *H* | −0.19 | −0.69 ** | 0.41 | −0.42 | −0.51 * | −0.06 | 0.08 | −0.01 | 1.00 | |
| $T_l$ | *C* | 0.31 | 0.46 | −0.19 | 0.21 | −0.18 | 0.68 | 0.89 ** | −0.87 ** | −0.09 | 1.00 |
| | *H* | −0.03 | 0.13 | −0.28 | 0.30 | −0.26 | 0.59 ** | 0.86 ** | −0.82 ** | −0.33 | 1.00 |

* Correlation is significant at the 0.05 level (2-tailed *t*-tests). ** Correlation is significant at the 0.01 level (2-tailed *t*-tests).

## 4. Discussion

### 4.1. The Limiting Factors Affecting Photosynthesis

Four major factors affect soil moisture regime in our studied system: precipitation, irrigation, crop evapotranspiration and soil evaporation. Soil moisture content under the same irrigation level was different in different seasons. In July, the soil moisture content within a 60–80 cm soil depth for the W2 treatment was lower than the W1 treatment. Plant roots are mainly concentrated in the 0–80 cm soil layer [25], and the plant transpiration in desert soil was below 60 cm, so these conditions may have resulted in the lower soil moisture contents in the W2 treatment. With the decrease of soil evaporation and plant transpiration in August and September, the soil water was replenished at the 0–80 cm depth. The irrigation levels should be adjusted by combining the environmental conditions, soil moisture content and water requirements of plants in different seasons.

Water is the primary limiting factor for plant growth in arid and semiarid regions. For desert plants, the compromise occurs in nature between restricting water loss through

the stomata versus maintaining a high carbon gain, which depends on both stomatal and non-stomatal regulations [3]. The intercellular $CO_2$ concentration ($C_i$) should change in the same direction as the $P_N$ if stomatal responses are dominant in response to perturbations to the assimilation rate. If the changes are opposite, the most crucial difference must have been in the mesophyll cells [26], caused by non-stomatal factors [7,18].

When plants are exposed to moderate or severe drought stress, the stomatal restriction is no longer the main reason for the decrease of photosynthesis and the role of non-stomatal restriction becomes more important [7]. In the present study, the changes in the $P_N$ of $C$ between 11:00 and 15:00 h in the W1 treatment and between 13:00 and 17:00 h in the W3 treatment were in the same direction as the intercellular $CO_2$ concentration ($C_i$) (Figure 6), which indicated that stomatal factors controlled the photosynthetic processes at these times. The $C_i$ changed in the opposite direction as the $P_N$ in the W2 treatment throughout the day, indicating that non-stomatal factors changed the photosynthetic processes. The $C_i$ changed in the same direction as the $P_N$ for $H$ in the W1 treatment between 13:00 and 17:00 h and the W3 treatment between 11:00 and 13:00 h, which indicated that stomatal factors changed the photosynthetic processes during this period. The photosynthetic processes of $C$ and $H$ in the W2 treatments were restricted to non-stomatal factors through the day and changes in soil moisture between 60–80 cm. These phenomena indicated that $P_N$ was mainly related to the soil water status in the main root system activity layer. The $P_N$ values for $C$ and $H$ were mainly limited by non-stomatal factors under severe water stress [27].

A "midday depression" is one of the ways for plants to adapt to a dry environment through evolution. $C$ and $H$ are typical desert plants and the diurnal variations of $P_N$ and $T_r$ for $C$ in the W1 treatment presented an obvious "midday depression" phenomenon, but the W2 and W3 treatments presented a slight "midday depression" phenomenon. This indicates that $C$ was under serious water stress, resulting in partial stomata closure to reduce transpiration. In contrast, the diurnal variations of $P_N$ for $H$ in the W2 treatment exhibited an obvious "midday depression" phenomenon because the curves of $P_N$ for $H$ in the W1 and W3 treatments were unimodal and there were no symptoms that $H$ suffered water stress in the W1 treatment.

In this study, the daily average values of $P_N$ and $T_r$ for $C$ were obviously higher than for $H$ under the same irrigation levels. The $C$ in the W1 treatment suffered from water stress, and $H$ in the W1 treatment did not demonstrate drought stress, which indicated that $H$ is more drought-tolerant than $C$, and a lower $T_r$ benefits water preservers.

Many studies have suggested that desert plants generally have high water use efficiency, especially under water stresses; high water use efficiency is usually an effective way to resist drought stress [28–30]. The daily water use efficiency (WUE) of $C$ was 1.61, 1.91 and 1.66 $\mu molCO_2 \ mmolH_2O^{-1}$ for the W1, W2 and W3 treatments and for $H$ it was 1.25, 2.20 and 1.24 $\mu molCO_2 \ mmolH_2O^{-1}$, respectively. For the two plant species, the WUE was higher in the W2 treatment. This indicated that light water stress could improve the WUE of $C$ and $H$, but WUE cannot be improved when plants are water deficient. This result may occur because water deficiency inhibits the plant's photosynthetic rate. Many stressful climate features include high temperature, high radiation and high evaporation in arid desert regions. The plant must maintain a certain transpiration rate to reduce the temperature and resist burns on its leaves [31].

Many environmental factors such as high light intensity, temperature and low soil moisture cause photo inhibition [32–34]. Our results indicated that environmental factors such as PPFD, $T_a$ and RH changed from morning to night, as indicated by the bivariate correlation test results comparing the environmental variables and plant physiology characteristics presented in Table 2. $g_s$ is a major determinant of the photosynthetic rate under well-watered conditions and non-stomatal limitations during drought conditions. $P_N$ and $T_r$ for $C$ and $H$ were significantly correlated with $g_s$. This result is consistent with previous findings [35–37]. The $P_N$ of $C$ was significantly correlated with PPFD, $C_i$, $L_s$ and $g_s$. However, the $P_N$ of $H$ was correlated with $g_s$. The WUE of $C$ was correlated with $P_N$, $C_i$ and $L_s$, while the WUE of $H$ was correlated with $T_r$ and $g_s$. All of the results showed that

the photosynthetic organs of *C* are more sensitive to the PPFD changes than *H*. *C* can adjust the relationship between respiration and photosynthesis, resulting in increases in $P_N$ and WUE. However, *H* can adjust the $g_s$ to decrease $T_r$, thus resulting in a higher WUE.

### 4.2. Plants' Adaptation Strategy to Drought Stress

The photosynthesis–light response curve helps estimate the adaptation of plants to their habitats. The light compensation point (LCP), light saturation point (LSP), apparent quantum yield of $CO_2$ assimilation (Φ), dark respiration rate ($R_d$) and light saturated rate of $CO_2$ assimilation ($P_{max}$) of *C* and *H* were different for the all of the irrigation treatments during the July, August and September throughout the experimental period. These physiological parameters reflect the adaptability of plants to environmental changes (Table 1).

The LCP reflects the ability of plants to use weak light, LSP demonstrates the ability of plants to use strong light and different values between LSP and LCP reproduce the plants' light adaptation. In the present study, the LCP of *C* and *H* in the W2 treatment was lower than the W1 and W3 treatments from July to August but was higher in September, and soil evaporation was stronger in July and August. Plants were stressed in two months. In September, soil moisture was higher, indicating that *C* and *H* can decrease LCP to adapt to drought stress [38]. The phenomenon of photosynthesis domestication appeared. The LSP of *C* was greater in the W3 treatment in July and August, but the LSP of *H* was not regular for the three irrigation levels. The different LSP and LCP values of *C* in the W3 treatment were higher from July to August. The values of *C* and *H* in W2 were lower than the W1 and W3 treatments in September, which indicated that *C* might make better use of strong light and show better light adaptation under better water conditions. Drought stress might improve the light adaptation of *H* and cause *H* to adapt to drought stresses.

$R_d$ for *C* and *H* was higher in the W3 treatment than in the W1 and W2 treatments during the three months we tested. This indicated that drought stress might slow the plants' metabolic capacity. This is a protective response of plants to drought stress. A decrease of $P_N$ under drought stress led to a shortage of photosynthetic matter in the body. A decrease of $R_d$ reduces the consumption of organic matter due to breathing, and it ensures that plants have a relatively large number of photosynthetic substances to construct competitive organs, such as the main root, lateral root, etc. Drought stress might relate the activity of key enzymes associated with changes in respiration concentration [39], and may also be caused by long-term severe drought stress destruction of plant cell structures related to respiration.

*C* in the W3 treatment had a higher $P_{max}$ than W1 and W2 treatments. However, *H* in the W3 treatment had a lower $P_{max}$ than the W1 and W2 treatments. The photosynthesis potential of *C* slowed down during drought stress, but drought inspired *H* photosynthesis potential. Furthermore, studies have shown that the plant shoots can improve their resistants (such as drought), making their newly developed photosynthesis potential as an effective means to increase yield [40–42]. The dark respiration rate for shoots of *H* is mainly affected by the effective photosynthetic radiation, air temperature, and relative humidity. However, photosynthesis rate is less affected by the environment, making its strong resistance [42]. At present, the contribution of shoot photosynthesis to drought tolerance cannot be ignored, and the response of shoot photosynthesis to drought needs to be further explored.

Finally, in addition to the two species selected in this study, several other species, particularly Tamarix ramosissima, are widely distributed in the region. As there are significant photosynthesis differences among plants and groundwater salinities, species combinations at different shelterbelt engineering areas differ. It is necessary to research more species with regard to plant species survival characteristics by using water-controlled experiments for a complete understanding of physiological mechanisms that is difficult to obtain in natural conditions due to the lack of a vast water range. More suitable plants should be selected, and sustainable irrigation technologies should be developed in this area and beyond.

## 5. Conclusions

This study demonstrated that two species (*C* and *H*) have salt resistance capabilities that maintain normal leaf scale photosynthesis during topsoil water stress conditions. Their adaptive strategies are eco-physiologically different at the species level. The soil water availability determines the plant response, acclimation and adaptation in this mobile sand desert. $P_N$ was mainly related to soil water status in the main root system activity layer. The daily variations of $P_N$ and $T_r$ for *C* were higher than that for *H* in July. Compared to *C*, *H* demonstrated a stronger capability for drought resistance. *C* is more sensitive to changes in PPFD than *H* as *C* increased WUE through increased $P_N$, but *H* decreased $T_r$ to obtain a higher WUE. Drought reduced the use of weak light and metabolic capacity of *C* and *H*, and decreased the light adaptability and photosynthesis potential of *C* instead of *H*. To evaluate the drought hardiness of plant species survival for extreme desert conditions, which are often difficult to obtain in natural conditions, our water-controlled studies are beneficial for a complete understanding of physiological mechanisms and possible plant morphological adjustments.

**Author Contributions:** Conceptualization and methodology, J.L. and Y.Z.; formal analysis, J.L.; investigation, J.L., H.L. and T.A.S.; resources, Y.W.; writing—original draft preparation, J.L.; writing—review and editing, J.L., Y.Z. and J.Z.; supervision, Y.Z. and J.Z.; project administration, Y.Z., Y.W. and J.Z. All authors have read and agreed to the published version of the manuscript.

**Funding:** This research was supported by the National Natural Science Foundation of China (41977009, 41877541, 41471222) and the National Talents Project (Y472241001).

**Conflicts of Interest:** The authors declare no conflict of interest.

## Abbreviations

AQE—apparent quantum yield; $C_i$—intercellular $CO_2$ concentration; $g_s$—stomatal conductance; k —the curved angle of the light response curve; $L_s$—stomatal limitations; LCP—light compensation point; LSP—light saturation point; RH—air relative humidity; $T_a$—air temperature; $T_l$—leaf temperature; $T_r$–transpiration; $P_N$—net photosynthetic rate; $P_{max}$—maximum net photosynthetic rate light; Q—effective photosynthetic radiation incident to the leaf; $R_d$—dark respiration rate; $\Phi$—apparent quantum yield; WUE—water use efficiency.

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
