# Peer review of "Photosynthetic Responses of Two Woody Halophyte Species to Saline Groundwater Irrigation in the Taklimakan Desert"

_water, doi:10.3390/w14091385_

Round 1
Reviewer 1 Report
The authors have presented a manuscript, which evaluated the photosynthetic response of two woody halophyte species in the Taklimakan desert. The manuscript presents interesting results concerning the selection and the response in the drought stress, but they are some point, which need to improve. Following, I have included some comments aimed to enhance the paper:
- The authors must include mores details and related studies in the introduction, it is very short. Improve the introduction and the presentation of the objectives.
- This work presents very interesting results. I think that the authors can improve the format of results demonstration. The authors can highlight better the importance of the results obtained.
· Two trees have been chosen (H. ammodendron and C. mongolicum) in the area and a series of individuals to study; I would like them to talk about some comparison with other species. · Consider extending the conclusions and adding a Future works paragraph.
Finally, the topic of this manuscript is interesting; since the selection of plants in the Taklimakan desert, but authors must restructure the manuscript to improve the readability of the text and their future trends and challenges.
Author Response
The authors have presented a manuscript, which evaluated the photosynthetic response of two woody halophyte species in the Taklimakan desert. The manuscript presents interesting results concerning the selection and the response in the drought stress, but they are some point, which need to improve. Following, I have included some comments aimed to enhance the paper:
Reply: Thank you for your pertinent and encouraging comments. Following your suggestion, we have improved our manuscript.
- The authors must include mores details and related studies in the introduction, it is very short. Improve the introduction and the presentation of the objectives.
Reply: Thanks for the pertinent suggestions. We have improved those points by adding more introductions. We also present our research objectives more clearly, like “Only when careful choices are made concerning the salinity of available irrigation water and the salinity tolerance of appropriate plant species can vegetation restoration promise sustainable development.”
- This work presents very interesting results. I think that the authors can improve the format of results demonstration. The authors can highlight better the importance of the results obtained.
Reply: Thank you for your pertinent and encouraging comments. Following your suggestion, we have improved the importance of the results.
- Two trees have been chosen (H. ammodendron and C. mongolicum) in the area and a series of individuals to study; I would like them to talk about some comparison with other species. · Consider extending the conclusions and adding a Future works paragraph.
Reply: Thanks for the pertinent suggestions. We have improved those points by adding one new paragraph at the end of the discussion. For example, “Finally, in addition to the two species selected in this study, several other species, particularly Tamarix ramosissima, are widely distributed in the region. Because there are significant photosynthesis differences among plants and groundwater salinities, species combinations at different shelterbelt engineering areas differ. It is necessary to research more species with regard to plant species survival characteristics by using water-controlled experiments for a complete understanding of physiological mechanisms that is difficult to obtain in natural conditions due to the lacking of a vast water range. More suitable plants should be selected, and sustainable irrigation technologies should be developed in this area and beyond.”
Finally, the topic of this manuscript is interesting; since the selection of plants in the Taklimakan desert, but authors must restructure the manuscript to improve the readability of the text and their future trends and challenges.
Reply: Thanks for the pertinent suggestions. We have improved those points by adding more content in the last-second paragraph of the discussion.
Reviewer 2 Report
The paper "Photosynthetic responses of two woody halophyte species to irrigated water in the Taklimakan Desert" presents an important topic.
The authors did a great job, however, some parts of the paper need to be improved before it can be accepted for publication.
- One English editing is needed to improve the readability of the work
- The references used are few and narrow the spectrum of the scientific soundness of the paper. I suggest consulting more related scientific works and broaden your discussion of the results in contrast. Add more references and cite them according to mdpi format.
- The studied area is always of interest to readers, please add a map of the studied area.
- The conclusion would be more interesting if you include the goal of the study.
Author Response
The authors did a great job, however, some parts of the paper need to be improved before it can be accepted for publication.
Reply: Thank you for your pertinent and encouraging comments. Following your suggestion, we have improved our manuscript.
- One English editing is needed to improve the readability of the work.
Reply: We have asked professor Robert Lee Hill from Maryland University for the language check.
- The references used are few and narrow the spectrum of the scientific soundness of the paper. I suggest consulting more related scientific works and broaden your discussion of the results in contrast. Add more references and cite them according to mdpi format.
Reply: Thanks for the pertinent suggestions. We have improved those points by adding one new paragraph at the end of the discussion (including 2-3 literature).
- The studied area is always of interest to readers, please add a map of the studied area.
Reply: We have added figure 1.
- The conclusion would be more interesting if you include the goal of the study.
Reply: We have improved as suggested.
Reviewer 3 Report
In the present manuscript, the daily dynamics of gas
exchange parameters and their responses to photosynthetic photon flux density at three irrigation levels were measured for Calligonum
mongolicum and Haloxylon ammodendron.
Lines 21-22: "but H. ammodendron" the sentence does not make sense.
Use italics for the Latin names of the species throughout the text.
Present EC values in dS/m.
Present the field capacity values of soil.
Add statistics in Fig 1 and 2. Otherwise, no terms that indicate significant differences can be used in the text.
Fig 3-5: Describe in the footnote what H and C mean.
Lines 233-236: this sentence should be placed in the Discussion section.
Table 1: add statistics.
Describe in the materials and methods how WUE was calculated.
Author Response
In the present manuscript, the daily dynamics of gas exchange parameters and their responses to photosynthetic photon flux density at three irrigation levels were measured for Calligonum mongolicum and Haloxylon ammodendron.
Reply: Thank you for your pertinent and encouraging comments. Following your suggestion, we have improved our manuscript.
- Lines 21-22: "but H. ammodendron" the sentence does not make sense.
Reply: "but H. ammodendron" has been replaced by "and H. ammodendron", and further modified into a new sentence.
2.Use italics for the Latin names of the species throughout the text.
Reply: We have improved those as suggested.
3.Present EC values in dS/m.
Reply: “EC” has been deleted in the revision.
4.Present the field capacity values of soil.
Reply: Field capacity value has been added in line 141.
5.Add statistics in Fig 1 and 2. Otherwise, no terms that indicate significant differences can be used in the text.
Reply: We have added the standard deviation value in the Figures, and also indicated the significant differences in the text.
6.Fig 3-5: Describe in the footnote what H and C mean.
Reply: Footnote has been added “H means H. ammodendron, C means C. mongolicum.”
7.Lines 233-236: this sentence should be placed in the Discussion section.
Reply: In the revision, this sentence has been deleted.
8.Table 1: add statistics.
Reply: We cannot add the statistical values since those measurements are only conducted once.
9.Describe in the materials and methods how WUE was calculated.
Reply: In the revision, the calculation of WUE has been added.
Round 2
Reviewer 3 Report
The authors addressed my comments.
Author Response
Thank you for your pertinent and encouraging comments.